# Investigating the relationship between internal spinal alignment and back shape in patients with scoliosis using PCdare: A comparative, reliability and validation study

Mirko Kaiser[1]*, Meby Mudavamkunnel[1], Martin Bertsch[1], Christoph J. Laux[2], Ines Unterfrauner[2], Florian Wanivenhaus[2], David E. Bauer[2], Thorsten Jentzsch[2], Alexandra Stauffer[2], Mazda Farshad[2], Sasa Cukovic[1]

1 Laboratory for Movement Biomechanics, ETH Zurich, Switzerland, 2 Balgrist University Hospital, University of Zurich, Switzerland

* mirko.kaiser@hest.ethz.ch

## Abstract

Optical 3D surface scanning has emerged as a valuable modality for assessing spinal deformity in patients with scoliosis, avoiding radiation exposure. However, correlations remain moderate between deformation parameters obtained from radiographs and those estimated solely from the 3D back surface, referred to as the "back-shape-to-spine" approach. To improve the accuracy with which the back-shape-to-spine approach can estimate the internal spinal alignment (ISL) from 3D surface scanning, deeper understanding is required of the effect of scoliosis on the back shape. The PCdare software, which enables semi-automatic registration of 3D surface scans with the corresponding biplanar radiographs, has been used by students in a previous study to validate study protocols, generate references for estimated ISL, and evaluate correlations between the spinous process line (SPL) and ISL. This study explored the potential of the PCdare software to investigate the underlying relationship between the ISL and the 3D back shape, conducted a comparative study with 3 study protocols, and conducted an inter- and intrarater reliability (IIR) study with 6 clinicians and 10 students as raters to evaluate the applicability of PCdare when used by students. The comparative study involved 252 patients with idiopathic scoliosis from 3 studies that compared the back-shape-to-spine approach with radiography. The quality of study protocols and the relationship between internal spinal alignment and 3D back shape were both investigated by evaluating the posture alignment errors and correlations between Cobb angles. The inter- and intrarater reliability study involved 7 patients with idiopathic scoliosis and was conducted using PCdare and validated with PACS. The median Cobb angle difference (interquartile range: IQR) between students and clinicians (interclass) was 0.06° (1.5°). The ICC [confidence interval] between Cobb angles (interrater) was 0.94 [0.7,0.98]. The median absolute

**Data availability statement:** All relevant data for this study are publicly available from the GitHub repository (https://github.com/mkaisereth/PCdareSoftware).

**Funding:** This research was funded by Innosuisse, grant number 47195.1 IP-LS. The funding was awarded to the institutions, not authors. The funder had no role in the study design, data collection and analysis, decision to publish, or preparation of the manuscript. https://www.innosuisse.admin.ch/en

**Competing interests:** The authors have declared that no competing interests exist.

Cobb angle difference (IQR) between 3 repetitions (intrarater) were 4.2° (5.3°) or lower. The median Cobb angle difference (IQR) between PCdare and PACS was 1.5° (8.4°) for clinicians and 1.4° (6.9°) for students, whereas the corresponding correlation [confidence interval] was 0.94 [0.92,0.96] and 0.95 [0.93,0.96], respectively. The median RMSE (median SD) of posture alignment error ranged between 8.1 mm (5.2 mm) and 5 mm (3.5 mm), whereas the median PCC (IQR) between ISL and SPL ranged between 0.64 (0.58) and 0.99 (0.02). Students achieve outcomes comparable to clinicians when using PCdare, which underlines its reliability and ease of use. In addition, the application of PCdare to examine the quality of study protocols revealed the necessity of markers and posture alignment and delivered correlation coefficients for the relationship between internal spinal alignment and 3D back shape. These findings highlight the potential of the PCdare software to advance the non-ionizing assessment of spinal deformities and thus improve understanding of scoliosis.

## Introduction

Adolescent idiopathic scoliosis (AIS) is by far the most common deformity of the spine, affecting up to 3% of teenagers globally [1]. If left untreated, AIS can induce cardiopulmonary impairment, cosmetic alterations, and discomfort [1]. The current clinical gold standard for assessing AIS is radiography. Several studies have reported that the incidence of radiation-induced cancer types in patients with AIS is 5 times higher than in individuals without AIS [2,3]. As a result, optical 3D surface scanning is increasingly investigated and applied medically as a non-ionizing screening method, mostly in young individuals [4–6]. However, correlations remain moderate between Cobb angles obtained from radiographs and those estimated solely from the 3D back surface, referred to as the "back-shape-to-spine" approach [4]. To improve the back-shape-to-spine approach and the algorithms that estimate internal spinal alignment, deeper understanding is required of the effect of scoliosis on the shape of the back.

Kaiser et al [7] investigated the relationship between internal spinal alignment and 3D back shape with a dataset of 30 patients with AIS using the PCdare software. The PCdare software can be used to register 3D surface scans with corresponding biplanar radiographs semi-automatically. Within PCdare, users manually select anatomical marker positions on both the 2D radiographs and the 3D surface scans and draw lines through the centroids of vertebral bodies (ISL) and along the spinous processes (SPL). The software then projects the 2D marker positions into 3D and computes their intersections. It then registers the 3D surface scans to the radiographic coordinate system using a rigid transformation computed via SVD-based least-squares fitting. The transformed SPL and ISL can then be used to calculate spatial correlations, which subsequently allows the relationship between the shape of the back and the internal alignment of the spine to be evaluated in patients with scoliosis. Their evaluation demonstrated time savings (requiring approximately 2 minutes compared to the 5–10 minutes typically required for the process), insights into the relationship between internal spinal alignment and 3D back shape, and the potential

for using PCdare to validate study protocols and internal spinal alignments (ISLs) estimated with the back-shape-to-spine approach.

Furthermore, students can use PCdare to reduce the workload for clinicians involved in such research projects [7]. However, the inter- and intrarater reliability of students using the PCdare software was not investigated. Hence, the aim of our study was to conduct an inter- and intrarater reliability study in which the PCdare software was used to draw lines through the centroids of the vertebral bodies on radiographs to identify the ISL. Furthermore, we have expanded the use of PCdare and applied it to datasets from 3 studies to provide more insights into the quality of the study protocols and ultimately more insights into the relationship between internal spinal alignment and 3D back shape.

## Materials and methods

The relationship between ISL and back shape was investigated with the PCdare software in three steps. First, an inter- and intrarater reliability (IIR) study confirmed the applicability of PCdare to generating reference lines and angles (first subsection). Second, the study protocols were validated by comparing the postural alignments identified by EOS imaging and by 3D scanning (second subsection). Third, characteristic properties of external shape were compared with internal alignment by examining the correlation between spinous process line (SPL) and ISL (third subsection).

The use of PCdare was evaluated with datasets from three studies, here numbered chronologically (Fig 1, Table 1). The first dataset contains optical 3D surface scans of 192 patients with AIS and was collected at an orthopedic institute

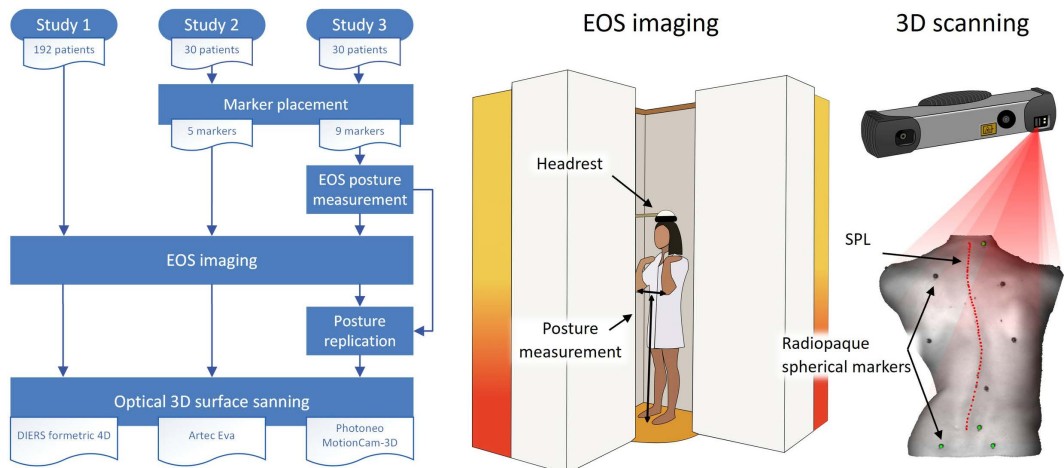

**Fig 1. Data collection procedure for all 3 studies (left).** EOS imaging (middle) and optical 3D surface scanning (right) was always performed; markers were only placed in Studies 2 and 3, and EOS posture measurement and replication was only performed for study 3.

**Table 1. Inclusion and exclusion criteria, demographic and anthropometric data for all three studies.**

|  | Study 1 | Study 2 | Study 3 |
|---|---|---|---|
| Number of patients | 192 (160 f, 32 m) | 30 (20 f, 10 m) | 30 (21 f, 9 m) |
| Inclusion criteria | 8-18 years<br>Cobb angle > 5° | 9-20 years | 8-26 years |
| Exclusion criteria | Non-idiopathic AIS<br>Obesity Pregnancy | Non-idiopathic AIS<br>Pregnancy | Non-idiopathic AIS<br>Pregnancy |
| Mean age ± SD | 13 ± 2 years | 13 ± 3 years | 18 ± 4 years |
| Mean Cobb angle ± SD | 33° ± 15° | 24° ± 12° | 28° ± 12° |

(Study 1) [4]. The second dataset contains scans of 30 patients with AIS and was collected at an academic spine center (Study 2) [8]. The third dataset contains scans of 30 patients with AIS and was collected at another academic spine center (Study 3). Study 3 was conducted between 12th December 2022 and 16th December 2024 in accordance with the Declaration of Helsinki and approved by the Cantonal Ethics Committee in Zurich (2022−01672, 22 November 2022). Written informed consent was obtained from all participants, for minors of age 14 or younger, written consent was obtained from parents or guardians.

The data collection procedure for all 3 studies was as follows (Fig 1, left). For Study 3, the patients were equipped with radiopaque spherical markers with a diameter of 5 mm on anatomical landmarks. The patients then underwent biplanar radiography (EOS imaging) in an upright standing posture as part of their standard medical treatment. The distance between elbows and floor and the distance between elbows was measured during EOS imaging to record posture (Fig 1, middle). Afterwards this posture was replicated while the patient's back was scanned with a Photoneo MotionCam-3D optical 3D scanner (Fig 1, right). For Study 2, the patients were also equipped with radiopaque spherical markers. The patients then underwent EOS imaging; however, the posture was not measured. Afterwards, the posture was replicated with the same instructions as during EOS imaging and the patient's back was scanned with an optical 3D scanner, an Artec EVA. For Study 1, no markers or posture measurement were used; the patients underwent EOS imaging and afterwards each patient's back was scanned using DIERS formetric 4D.

## Method of inter- and intrarater reliability

First, an a priori power analysis (outlined later in this section) was conducted using R (RStudio 2023.12.1 Build 402) to estimate the required number of clinicians, students, selected images and repetitions. These estimates were then adjusted based on practical considerations, such as clinician availability and workload. In total, 6 clinicians, who were neurosurgeons and orthopedic surgeons, and 10 health science and technology students participated in the inter- and intrarater reliability (IIR) study (Fig 2). The clinicians were recruited at an academic spine center and the students at a

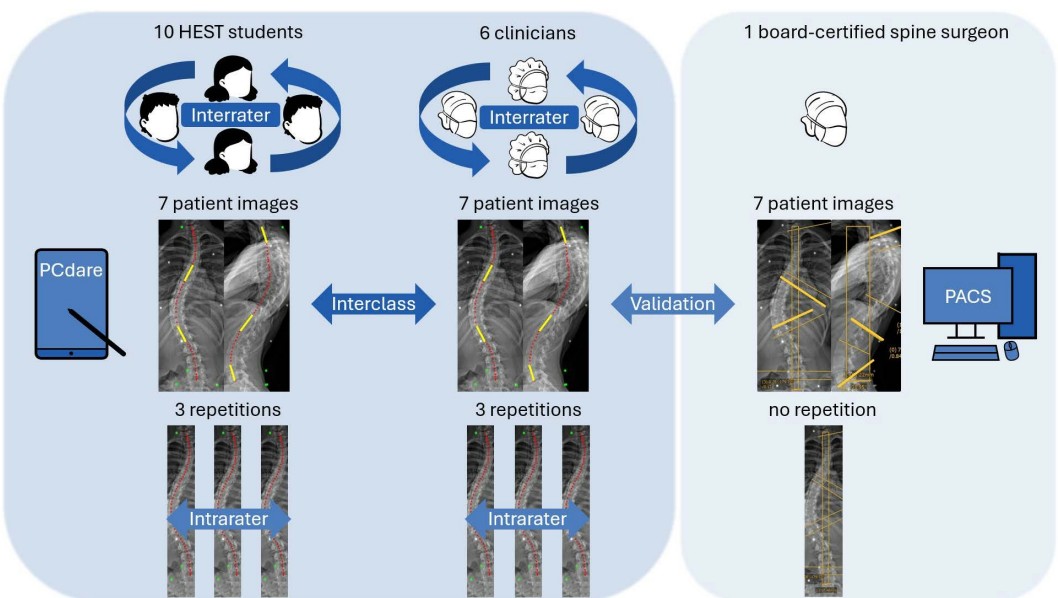

**Fig 2. Inter- and intrarater reliability (IIR) study: Interclass reliability between clinicians and students, inter- and intrarater reliability among clinicians and among students, and validation with manual annotation in PACS by a board-certified spine surgeon.**

laboratory for movement biomechanics. The clinicians and the students each used the PCdare software in single sessions to draw the ISLs on two 2D biplanar radiographic images for 7 pseudo-randomly selected patients from the dataset of Study 3. This dataset was chosen due to its superior data quality compared to Studies 1 and 2, particularly in terms of smaller posture alignment errors (second subsection in Results). The selected patients reflected the AIS severity distribution of Study 3, therefore, the 7 selected images were from 1 patient with a normally shaped spine (Cobb angle less than 10°), 2 patients with mild scoliosis (Cobb angle less than 25°), 3 patients with moderate scoliosis (Cobb angle less than 45°) and 1 patient with severe scoliosis (Cobb angle of 45° or greater). Each patient image was repeated 3 times, and all images were then shuffled randomly, resulting in a total of 21 randomly presented biplanar images. Afterwards, the drawn ISLs were smoothed, and the Cobb angles of the coronal, kyphotic, and lordotic curvatures were calculated [7]. In addition, a board-certified spine surgeon annotated all 7 patient images in PACS (MERLIN Diagnostic Workcenter, Version 7.1, Phönix-PACS GmbH, Freiburg im Breisgau, Germany) as part of the medical routine. The Cobb method was used for coronal (endplates of most tilted vertebrae), kyphotic (T1–T12 endplates), and lordotic (L1–S1 endplates) curvature angles.

All subsequent statistical evaluations were performed in RStudio, and the associated R code is available on GitHub (see Data Availability Statement).The Cobb angles from PCdare were compared with the Cobb angles manually annotated in PACS (Fig 2, Validation) once by calculating Pearson correlation coefficients and once by calculating the median Cobb angle differences and interquartile ranges.

The interclass reliability between clinicians and students (Fig 2, Interclass) was calculated with the 90% confidence interval of *isClin* in a linear mixed-effects model (Equation 1),

$$\mathrm{fm_{CAE}} = \mathrm{lmer}\left(\mathrm{error} \sim \mathrm{isClin} + (1|\mathrm{raterID}), \ \mathrm{data}\right), \tag{1}$$

where *data* is the coronal Cobb angle ratings including *error*s, *raterID* is a unique ID for each rater, *isClin* is 1 for clinicians, 0 for students, and $\mathrm{fm_{CAE}}$ is the linear mixed-effects model for the coronal Cobb angle errors, fitted using the lmer() function from the lme4 package. Furthermore, the time required by students and clinicians was examined with time stamps. A Q-Q plot was used to verify the normal distribution of the errors of the coronal Cobb angles between PCdare and PACS.

The interrater reliability among clinicians and students (Fig 2, Interrater) was calculated with the intraclass correlation coefficient (ICC, Equation 2) [9],

$$\mathrm{fm_{CA}} = \mathrm{lmer}\left(\mathrm{cobbAngle} \sim \mathrm{isClin} + (1|\mathrm{patID}) + (1|\mathrm{raterID}), \ \mathrm{data}\right), \mathrm{ICC_{CA}} = \mathrm{performance::icc}\left(\mathrm{fm_{CA}}, \ \mathrm{ci} = \mathrm{TRUE}\right), \tag{2}$$

where *patID* contains a unique ID for each patient, $\mathrm{fm_{CA}}$ is the linear mixed-effects model for the coronal Cobb angles and *performance* :: *icc* (icc() function from the performance package) computes the ICC for the random effects in $\mathrm{fm_{CA}}$.

The intrarater reliability within the 3 repetitions of clinicians and students (Fig 2, Intrarater) was calculated once with the root mean square error (RMSE), standard deviation (SD), and interquartile range (IQR) between smoothed ISL lines and once with the absolute difference and IQR between coronal Cobb angles. Intraclass correlation coefficients were calculated separately for all 3 repetitions of each rater (Equation 3).

$$\mathrm{ICC2_{CA}} = \mathrm{psych::ICC(cobbAnglesRaterReps)}, \tag{3}$$

where psych:ICC (icc() function from psych package) calculates the ICC2 (2-way mixed effects, single measurement and absolute agreement) of *cobbAnglesRaterReps*, which represents all 3 repetitions of all 7 ratings for a single rater.

An a priori power and sensitivity analysis was performed in R using 16 raters, 7 images, 3 repetitions, an average coronal Cobb angle of 30°, and standard deviations of effect sizes for the patient effect $pat_{SD}$ from 3° to 6.7°, for rater effect

$rater_{SD}$ from 1.2° to 1.6°, for the repetition effect $rep_{SD}$ from 1.2° to 1.6°, and estimated difference $\hat{\beta}$ between students and clinicians from 1.5° to 3° (Equation 4),

$$\text{cobbAngle} = \mu + \hat{\beta} \cdot \text{isClin} + \text{pat}_{\text{effect}} + \text{rater}_{\text{effect}} + \text{rep}_{\text{effect}},$$
$$\text{fm}_{CA} = \text{lmer}\left(\text{cobbAngle} \sim \text{isClin} + \text{patID} + \text{repID} + (1|\text{raterID}),\ \text{data}\right),$$

(4)

where $\mu$ is the average coronal Cobb angle, $\hat{\beta}$ is the estimated difference between coronal Cobb angle ratings from students and clinicians, $\text{pat}_{\text{effect}} \sim N(0, \text{pat}_{SD}{}^2)$ is the patient effect, which is assumed to be normally distributed with mean 0 and standard deviation $\text{pat}_{SD}$, $\text{rater}_{\text{effect}} \sim N(0, \text{rater}_{SD}{}^2)$ is the rater effect, $\text{rep}_{\text{effect}} \sim N(0, \text{rep}_{SD}{}^2)$ the repetition effect, and *data* the simulated data. Equation (4) was simulated 500 times, and the power was calculated by checking how many times the 90% confidence interval of $\hat{\beta}$ was within [−5°,5°].

## Method of posture alignment

To validate the posture replication during Study 3 (Fig 1), PCdare was configured to automatically calculate the posture alignment error between radiographic images and 3D surface scans [7]. First, the lateral outlines (side view) of the back shape were located (Fig 3). Then, the root mean square error (RMSE) and standard deviation (SD) between both lateral outlines were calculated. The RMSE and SD were then used to validate the posture replication between the posture as captured by EOS imaging and the posture as captured by optical scanning.

## Method of correlating between SPL and ISL

We focused the investigation of the relationship between internal spinal alignment and external back shape on the correlation between SPL and ISL to confirm whether the ISL can be estimated with a linear model (Fig 4). The PCdare software was configured to automatically calculate the correlations between smoothed SPL and ISL for all 3 studies with 2 options [7]. Option 1: The Pearson correlation coefficient is calculated directly on SPL and ISL separately for the sagittal and

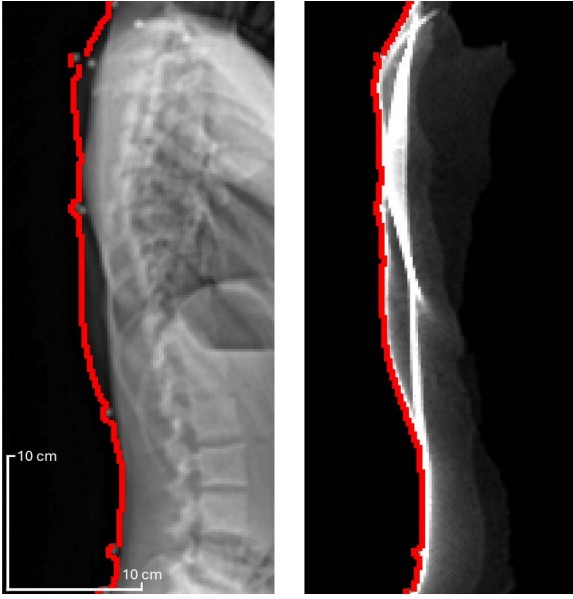

**Fig 3. Lateral outline (red) for radiographic image (left) and lateral projection of optical back surface scan (right).**

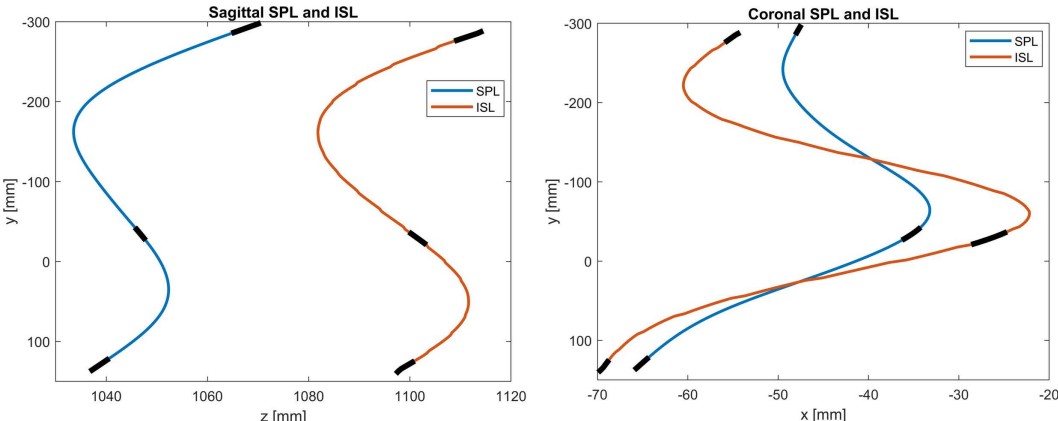

**Fig 4. Example of smoothed SPL and ISL for sagittal (left image) and coronal (right image) planes with indication of the positions of C7, T12, and L5 (black).**

coronal planes. Option 2: A Procrustes transformation is applied to the ISL before calculating the correlation coefficient in the same manner. The Procrustes transformation is configured to include only translation and rotation without scaling. This configuration allows the correlation of the shapes to be calculated irrespective of vertical translation and global rotation.

## Results

### Inter- and intrarater reliability

The Q-Q plot for 336 coronal Cobb angle errors from the IIR study confirmed the assumption of a normal distribution of the coronal Cobb angle errors. The power and sensitivity analysis (Equation 4) resulted in a power between 50% and 94% using 16 raters, 7 images, and 3 repetitions.

### Time requirements

During the IIR study, on average the clinicians needed 53 seconds with a standard deviation (SD) of 4 seconds to draw both line markings on both radiographic images of a single patient. For the same task, on average the students needed 74 seconds with an SD of 16 seconds.

### Intrarater and interclass reliability for RMSE between smoothed ISL lines

The median RMSE and SD between the smoothed ISL lines from the 3 repetitions from a single rater (Intrarater, Fig 2) for all 6 clinicians and for all 10 students were 2.0 mm or lower (Table 2).

**Table 2. Median RMSE and SD between the smoothed ISL lines from the 3 repetitions for all 6 clinicians and all 10 students.**

|  | Median RMSE (IQR) | Median SD (IQR) |
|---|---|---|
| Clinicians (intrarater) | 2.0 mm (1.2 mm) | 1.5 mm (0.8 mm) |
| Students (intrarater) | 2.0 mm (1.2 mm) | 1.5 mm (0.9 mm) |
| Difference (interclass) | 0.05 mm (0.03 mm) | 0.003 mm (0.06 mm) |

## Validation by correlation between Cobb angles calculated from smoothed ISL lines and manually annotated

The correlations of the coronal Cobb angles calculated from the smoothed ISL lines with the coronal Cobb angles manually annotated by a board-certified spine surgeon (Validation, Fig 2) were excellent (>0.9 [10]) for the ISL lines drawn by both clinicians and students (Table 3). The correlations of the kyphotic Cobb angles were very strong (>0.8) for the ISL lines drawn by both clinicians and students, and the correlations of the lordotic Cobb angles were moderate (>0.6) for the ISL lines drawn by the clinicians and strong (>0.7) for the ISL lines drawn by the students.

## Inter- and intrarater reliability and validation for Cobb angles calculated from smoothed ISL lines

**Interclass reliability and validation.** The median differences between clinicians and students for Cobb angles of coronal, kyphotic, and lordotic curvatures calculated from the smoothed ISL lines (Fig 2, Interclass) range between 0.06° and 0.4° (Table 4, Fig 5).

The 90% confidence interval for the difference in coronal Cobb angles using PCdare between clinicians and students (Interclass, Equation 1) was [−2.2°,1.9°] and therefore significant with a lower and upper limit of [−5°,5°].

**Table 3. Correlation [95% confidence interval] of Cobb angles of coronal, kyphotic, and lordotic curvatures calculated from smoothed ISL lines drawn by clinicians and students and manually annotated in PACS by a board-certified spine surgeon.**

| Cobb angle correlation [conf int] | Coronal | Kyphotic | Lordotic |
|---|---|---|---|
| Clinicians (PCdare) – PACS | 0.94 [0.92,0.96] | 0.89 [0.85,0.93] | 0.69 [0.58,0.77] |
| Students (PCdare) – PACS | 0.95 [0.93,0.96] | 0.90 [0.87,0.92] | 0.78 [0.72,0.82] |

**Table 4. Median differences (IQR) between clinicians and students for Cobb angles of coronal, kyphotic, and lordotic curvatures calculated from the smoothed ISL lines and median differences between Cobb angles calculated using PCdare and manually annotated by a board-certified spine surgeon in PACS.**

| Median Cobb angle difference (IQR) | Coronal | Kyphotic | Lordotic |
|---|---|---|---|
| Clinicians (PCdare) – PACS (Validation) | −1.5° (8.4°) | 3.9° (12.7°) | −19° (16.2°) |
| Students (PCdare) – PACS (Validation) | −1.4° (6.9°) | 4.3° (7.9°) | −20° (10.9°) |
| Students – Clinicians (Interclass) | 0.06° (1.5°) | 0.4° (4.9°) | 0.1° (5.3°) |

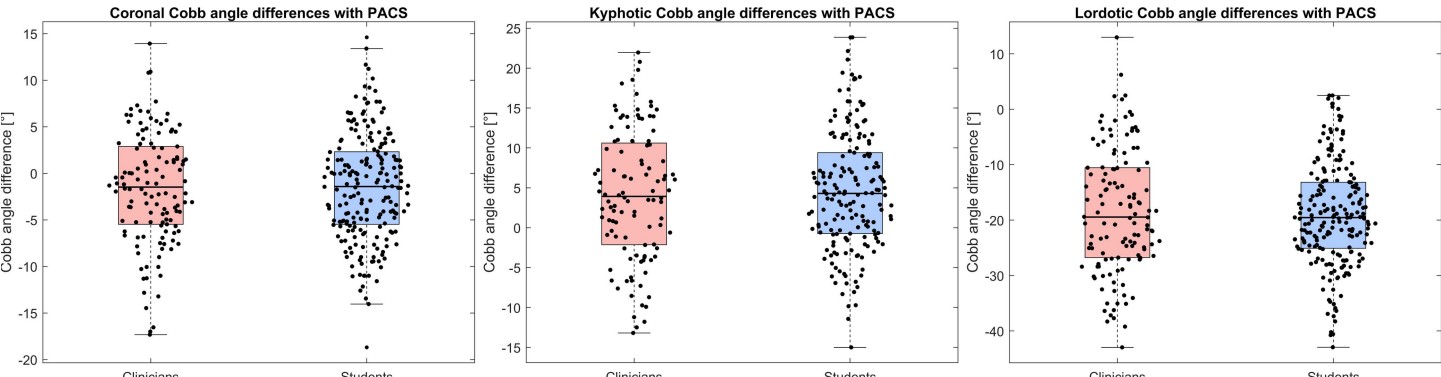

**Fig 5. Difference between coronal Cobb angles (left), kyphotic Cobb angles (middle), and lordotic Cobb angles (right) calculated from smoothed ISL lines drawn by clinicians and students (interclass) and manually annotated by a board-certified spine surgeon in PACS (validation).**

**Interrater reliability.** The ICC (Interrater, Equation 2) of coronal Cobb angle ratings was 0.94 with 95% confidence interval of [0.7,0.98].

**Intrarater reliability.** The median absolute difference and IQR between the coronal Cobb angles from the 3 repetitions from a single rater (Intrarater, Fig 2) for all 6 clinicians and for all 10 students were 5.3° or lower (Table 5).

The median ICC (Intrarater, Equation 3) for all 7 ratings, all 3 repetitions, and all raters was 0.95 [0.83, 0.99] with IQRs of 0.03 [0.08, 0.006].

## Posture alignment error

The median RMSE (median SD) between the lateral outline of the optical back surface scan and the lateral outline of the radiographic image for all patients in the datasets from Studies 1,2, and 3 were 8.1 mm (5.2 mm), 8.3 mm (4.7 mm), and 5 mm (3.5 mm), respectively (Fig 6).

## Correlations between SPL and ISL

The median Pearson correlation coefficient (PCC) and its interquartile range (IQR) for the correlation between smoothed SPL and ISL for all 3 datasets, for both sagittal and coronal planes and for both original smoothed line markings and smoothed line markings after applying a Procrustes transformation ranged between 0.64 (0.58) and 0.99 (0.02) (Table 6, Fig 7).

## Discussion

The IIR study with 6 clinicians and 10 students showed that the PCdare software is fast to use, that no special training is needed, and that clinicians and students can draw ISL lines very reliably. The median RMSEs between smoothed

**Table 5. Median absolute difference (MAD) and IQR between the coronal Cobb angles from the 3 repetitions from a single rater for all 6 clinicians and all 10 students.**

|  | MAD | IQR |
|---|---|---|
| Clinicians (intrarater) | 4.2° | 5.3° |
| Students (intrarater) | 4.0° | 5.1° |

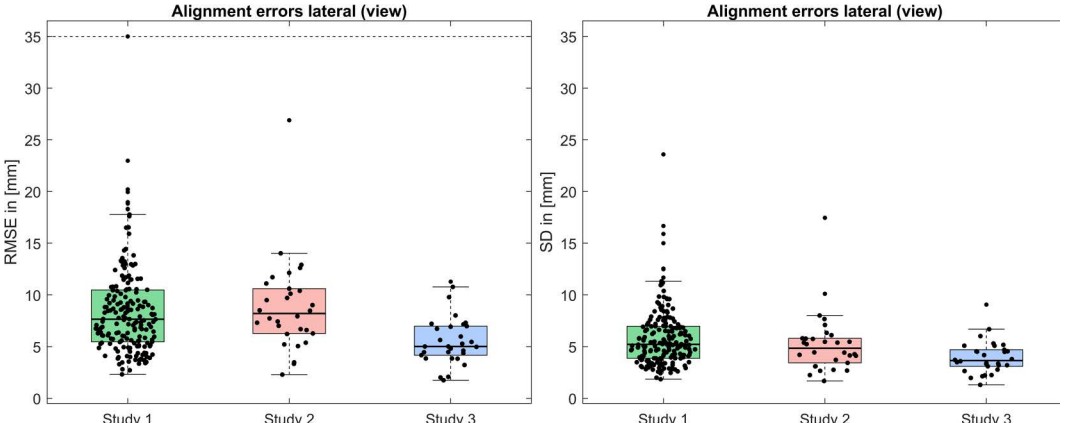

**Fig 6. Alignment errors: RMSE (left image) and SD (right image) between lateral outline of the optical back surface scan and lateral outline of the radiographic image for all patients in the datasets of Studies 1,2, and 3. The dashed line indicates that an outlier was visually truncated (original value: 36.3 mm).**

**Table 6. Median Pearson correlation coefficient (PCC) and interquartile range (IQR) for all 3 datasets from Studies 1,2, and 3. Each PCC is separately calculated for the sagittal and coronal plane, and for the original smoothed line markings of SPL and ISL and the smoothed line markings after applying a Procrustes transformation to the ISL.**

| Median PCC (IQR) | Study 1 | | Study 2 | | Study 3 | |
|---|---|---|---|---|---|---|
| | Sagittal | Coronal | Sagittal | Coronal | Sagittal | Coronal |
| Original SPL & ISL | 0.67 (0.57) | 0.64 (0.58) | 0.98 (0.05) | 0.75 (0.53) | 0.98 (0.04) | 0.79 (0.50) |
| Procrustes SPL & ISL | 0.77 (0.26) | 0.77 (0.29) | 0.99 (0.02) | 0.81 (0.41) | 0.99 (0.03) | 0.83 (0.33) |

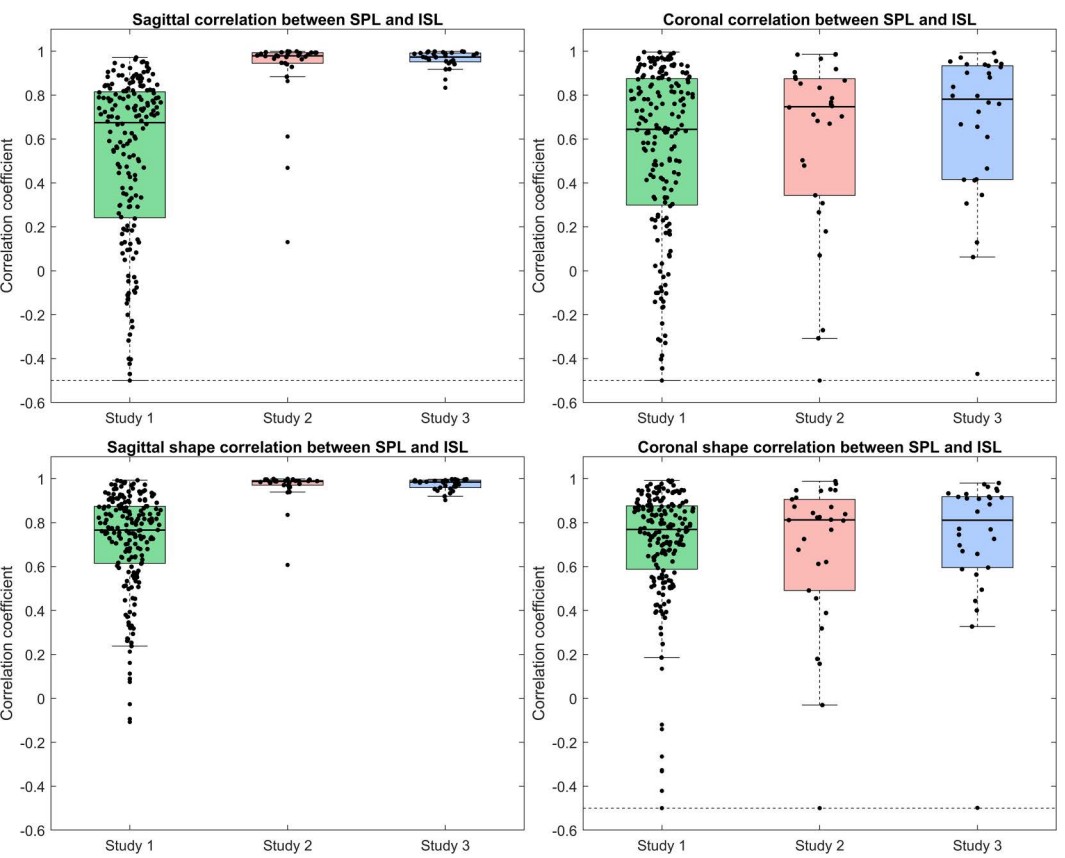

**Fig 7. Correlation between smoothed SPL and ISL lines for the sagittal plane (top left) and the coronal plane (top right); Shape correlation: after applying a Procrustes transformation, for the sagittal plane (bottom left) and the coronal plane (bottom right).** The dashed lines indicate that outliers were visually truncated (min value: −0.94).

ISL lines show that both clinicians and students can draw the ISL repeatedly with little variation. Because PACS does not provide the ISL, no reference ISL was available. Therefore, RMSE values were calculated between ISL drawings with PCdare. The Cobb angles calculated from the smoothed ISL had moderate to excellent correlation with the manual annotation in PACS. The coronal Cobb angle in particular had an excellent correlation for clinicians and students. In the literature, authors have reported Pearson correlation coefficients from 0.92 to 0.99 between coronal Cobb angles annotated manually in PACS [12]. The correlation between kyphotic Cobb angles was strong and between lordotic Cobb angles moderate. The contour of vertebral bodies is less clearly visible on lateral radiographs from EOS low-dose imaging, especially in the thoracic region, which could explain the lower correlation. One source of error for the lordotic Cobb angle

is the difference in definition of the angles. In PACS, the lordotic Cobb angle is annotated from L1 to S1 (upper endplate), whereas in PCdare the ISL is drawn until the lower endplate of L5. Another source of error is the influence of the smoothing of the ISL at the beginning and ending of the line.

Median differences between coronal Cobb angles derived using PCdare and PACS together with the 90% confidence interval show that our findings align with values reported in literature. In the literature authors have reported values between 1.0° and 7.2° [13–21]. Unfortunately, the authors reported their findings with heterogeneous metrics, which complicates the comparison. As a result, we defined a lower and upper limit of [−5°, 5°] for the equivalence test. The ICC for coronal Cobb angles with 95% confidence interval is also comparable to published values. In the literature, authors have reported ICC values between 0.89 and 0.99 [12,14,19,22].

The median time requirements for drawing the ISL on both radiographs suggest considerable time savings compared with manual annotation in PACS. In literature, authors have reported values between 5 and 20 minutes [23–25].

The PCdare software has been used to investigate the study protocols of 3 studies that examined the relationship between internal spinal alignment and back shape. Because optical scanning cannot be used simultaneously with EOS imaging, the images must be acquired sequentially. Furthermore, the EOS imaging room in hospitals is usually compact and frequently occupied; therefore, the patient must be transferred to another room for optical scanning. Replicating the same posture during optical scanning is therefore an important aspect of the study protocol. Evaluating the relationship between internal spinal alignment and back shape requires that posture alignment errors be minimized. The results showed that simply measuring the distance between elbows and floor and distance between elbows reduces the alignment error by 40%. Consequently, the posture alignment error for Study 3 is small, but the resulting uncertainty for the clinical aspects should be further investigated, including potential improvements to the study protocol to further reduce such errors.

The median Pearson correlation coefficients between SPL and ISL for the sagittal plane ranged from moderate for Study 1 to excellent for Studies 2 and 3. For the sagittal plane, a high correlation coefficient is to be expected even in patients with scoliosis, because the sagittal profiles of ISL and SPL are connected through the spinous processes and are less affected by vertebral rotation. The posture alignment error alone does not explain the differences in findings between the 3 studies. For example, Study 1 and Study 2 had similar rates of posture alignment error; however, Study 2 also used markers for registration between radiographs and optical surface scans, which could explain its high correlation. Therefore, the authors recommend combining posture measurements with markers to reduce sources of error. The median PCC for the coronal plane ranged from moderate for Study 1 to strong for Study 3. The lower correlation coefficients for the coronal plane can be explained by scoliosis. The deformation of the spine is usually accompanied by vertebral rotations around the craniocaudal axis.

Future work may explore several directions, including comparisons with ISLs extracted using other tools such as Surgimap [11], further improvements of the calculation of Cobb angles for coronal, kyphotic, and lordotic curvatures, and further automation of PCdare to increase time savings. Additionally, sources of error for the lordotic Cobb angle will be examined, and the use of PCdare to investigate vertebral rotations will be pursued.

Future work also includes assessing test-retest reliability by comparing results between sessions. During the IIR study, students completed a second session within a week of the first session. The data for assessing test-retest has already been collected but not yet evaluated. Furthermore, the authors recommend examining the test-retest reliability of posture alignment error. Expanding the IIR study to involve a larger number of clinicians using PACS would offer greater insight into the reliability of the validation. Moreover, repeating the IIR study with refined effect size estimates derived from the results of the IIR study presented here would also reduce power dispersion in the sensitivity analysis.

## Conclusion

We presented an inter- and intrarater reliability study, an investigation of 3 study protocols, and correlations between SPL and ISL with the PCdare software. The IIR study showed that PCdare does not require special training and that both

clinicians and students can draw ISL lines very fast and reliably. Validation with the current gold standard in clinical practice showed that coronal Cobb angles calculated with PCdare had excellent correlation for both clinicians and students, with median differences and standard deviations comparable to other published studies. Median differences for sagittal Cobb angles were systematically larger and are the focus of future work.

The evaluation of the protocols of 3 studies investigating the relationship between internal spinal alignment and back shape showed that measuring the posture during EOS imaging reduced posture alignment errors significantly. In Study 3, the correlations between SPL and ISL ranged from very strong to excellent, whereas in Study 1, the correlations ranged from moderate to strong. Therefore, the authors recommend the use of posture measurements in all future studies to improve data quality.

This work demonstrates that the PCdare software, which is publicly available on GitHub [7], is a reliable choice for drawing reference lines for SPL and ISL, for examining the quality of study protocols, and for investigating the relationship between internal spinal alignment and back shape. PCdare can help the scientific community to improve its understanding of the relationship between internal spinal alignment and back shape and therefore improve back-shape-to-spine approaches.

## Acknowledgments

The authors would like to thank Sabrina Catanzaro and Philipp Gähwiler for their outstanding support during the study at the Balgrist University Hospital; Volker M. Koch, Tobia Brusa, and Marco Wyss for supervision and their professional support with the optical scanning systems; Jasmin Wandel for her advice on statistics; Daniel Studer and Christoph Heidt for their clinical support during the study at the University Children's Hospital of Basel; Tito Bassani for the evaluation of the study data from the IRCCS Istituto Ortopedico Galeazzi, Milan, Italy; and Simon Milligan for his excellent language editing and advice.

## Author contributions

**Conceptualization:** Mirko Kaiser.

**Data curation:** Mirko Kaiser, Meby Mudavamkunnel, Martin Bertsch, Christoph J. Laux, Ines Unterfrauner, Florian Wanivenhaus, David E. Bauer, Thorsten Jentzsch, Alexandra Stauffer, Mazda Farshad, Sasa Cukovic.

**Formal analysis:** Mirko Kaiser, Meby Mudavamkunnel.

**Investigation:** Mirko Kaiser, Meby Mudavamkunnel, Martin Bertsch, Christoph J. Laux.

**Methodology:** Mirko Kaiser, Meby Mudavamkunnel.

**Resources:** Mirko Kaiser.

**Software:** Mirko Kaiser.

**Supervision:** Christoph J. Laux, Mazda Farshad, Sasa Cukovic.

**Validation:** Mirko Kaiser.

**Visualization:** Mirko Kaiser.

**Writing – original draft:** Mirko Kaiser.

**Writing – review & editing:** Mirko Kaiser, Meby Mudavamkunnel, Martin Bertsch, Christoph J. Laux, Ines Unterfrauner, Florian Wanivenhaus, David E. Bauer, Thorsten Jentzsch, Alexandra Stauffer, Mazda Farshad, Sasa Cukovic.

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
