## [Decision Letter · Decision Letter 0]

PONE-D-25-09339Investigating the relationship between internal spinal alignment and back shape in patients with scoliosis using PCdare: a comparative, reliability and validation studyPLOS ONE

Dear Dr. Kaiser,

Thank you for submitting your manuscript to PLOS ONE. After careful consideration, we feel that it has merit but does not fully meet PLOS ONE’s publication criteria as it currently stands. Therefore, we invite you to submit a revised version of the manuscript that addresses the points raised during the review process.

We look forward to receiving your revised manuscript.

Kind regards,

Mohammad Amin Fraiwan

Academic Editor

PLOS ONE

Additional Editor Comments (if provided):

Reviewers' comments:

Reviewer's Responses to Questions

**Comments to the Author**

1. Is the manuscript technically sound, and do the data support the conclusions?

Reviewer #1: Yes

Reviewer #2: Yes

Reviewer #3: Yes

2. Has the statistical analysis been performed appropriately and rigorously? 

Reviewer #1: No

Reviewer #2: Yes

Reviewer #3: Yes

3. Have the authors made all data underlying the findings in their manuscript fully available?

Reviewer #1: Yes

Reviewer #2: No

Reviewer #3: Yes

4. Is the manuscript presented in an intelligible fashion and written in standard English?

Reviewer #1: Yes

Reviewer #2: Yes

Reviewer #3: Yes

5. Review Comments to the Author

Reviewer #1: The manuscript written by Kaiser et al focuses on using PCdare to improve ease of interpretation for AIS. Overall, the clinical involvement and analysis is interesting, and the research aligns well with the guidelines in PLOS One. However, some essential revisions are needed before it is ready for publication.

1. It would be nice to have a reference supporting the claim that up to 3% of teenagers are affected by AIS (Line 58). Is this in the United States or globally? I specification on this would be helpful for context of AIS cases. Likewise, the authors mention that several studies have reported the incidence of radiation-induced cancer types is elevated in patients with AIS but only cites one reference. Is there more evidence to support the wording of “several”?

2. The PCdare software and the overview of what the software does should be explained in the introduction where Kaiser et al is mentioned. Perhaps a brief description of the software should be made after the sentence describing how the software can be used to register 3D surface scans on Line 73. This will really make the method and significance of this software stand out.

3. In regard to the time savings mentioned on Line 73, how much time was saved using such method? Having this specified may help show the significance of this approach compared to traditional methods.

4. The section written between Lines 101-112 is written out of order in respect to the content in Fig 1. It would naturally flow and read easier with the figure if Study 1 was mentioned first, Study 2, then to Study 3.

5. The authors mentioned on Line 123 that 7 images were used. Among them, 1 was with a normal spine, 2 with mild scoliosis, 3 with moderate, and 1 with severe. Was there a reason for the number of 7 images? And likewise, why were the patient breakdown unequal?

6. Were there considerations for statistical pairwise comparisons to generate a p value across students and clinicians to show if there is/lack of significance between the two groups for more informative results?

7. It appears that the equations written are in coding notation. If able, it would be more legible to present such calculations with the mathematical formulas, and referencing the packages/software used if any. I notice R is mentioned on Line 160, but not prior, which may be associated with the confusion.

8. Scale bar for the images in Fig 3. is missing.

9. Fig 5,6, and 7 should have all measurements plotted on the box plot to make it clear to the readers the variation of each reading, rather than the outliers only. It would further inform the type of distributions and clusters that are hidden in this current format.

10. Minor aesthetics comments on Fig 6 and 7. It appears Fig 6 is a screenshot, with the plot (left one) appearing to be clicked on prior to saving. Between these two figures, I see a minor dashed line in some plots, but not in others. What does this mean?

11. On Line 315-316, the authors mention how “a high correlation coefficient is to be expected even in patients with scoliosis.” Perhaps another sentence explicitly stating the reasoning may be informative to readers not fully aware of the field.

12. There are numerous areas that mention future work in the Discussion. (e.g., Lines 278, 290, 297, 301, 324, then the final paragraph in the Discussion). It may be easier to read if all mentions of future work is compartmentalized into one paragraph, perhaps at the end of the Discussion section into the paragraph along Lines 325-331.

Reviewer #2: The authors highlighted the PCdare research software usage through 3 experimental studies, and investigated its reliability as a drawing tool for internal spinal alignment(ISL) in this current work. The experimental setup was well explained for the all 3 studies and it managed to help readers understand the effect of the markers and posture measurement. The approach, methodology and results support the objective of the work. The discussion also proposes explanations for the encountered errors with possible expansions in future work.

It should be noted that code is made available but dataset is not shared but promised to be shared in a future data publication.

There are few points that may help readers understand the reasoning behind some of the steps taken in the analysis, the first of which is the reasoning for the selection of 7 images only from the study 3 with 30 subjects available?, another followup explanation is for the reason for the class distribution [1,2,3,1]? is this a stratified random selection? or the two extreme cases (normal and severe) are easily examined and drawing of ISL poses no challenge for example?, it is not clear as to why such distribution is made assuming all 30 subjects have AIS. The use of median can be understood after examining the box plots but it would be clearer to readers if it is explained to understand the justification.

Also, it might help if the reasons why images from study 1 and 2 are not used in the inter/intrarater reliability testing are explained, especially study 2 as it shows similar performance when compared to study 3(Table 5).

Some extra details for the study protocol, what are the selection criteria for the participants and annotator?, would additional annotators provide better understanding of variability?

Reviewer #3: This is an automated report for PONE-D-25-09339. This report was solicited by the PLOS One editorial team and provided by ScreenIT.

ScreenIT is an independent group of scientists developing automated tools that analyze academic papers. A set of automated tools screened your submitted manuscript and provided the report below. Each tool was created by your academic colleagues with the goal of helping authors. The tools look for factors that are important for transparency, rigor and reproducibility, and we hope that the report might help you to improve reporting in your manuscript. Within the report you will find links to more information about the items that the tools check. These links include helpful papers, websites, or videos that explain why the item is important. While our screening tools aim to improve and maintain quality standards they may, on occasion, miss nuances specific to your study type or flag something incorrectly. Each tool has limitations that are described on the ScreenIT website. The tools screen the main file for the paper; they are not able to screen supplements stored in separate files. Please note that the Academic Editor had access to these comments while making a decision on your manuscript. The Academic Editor may ask that issues flagged in this report be addressed. If you would like to provide feedback on the ScreenIT tool, please email the team at ScreenIt@bih-charite.de. If you have questions or concerns about the review process, please contact the PLOS One office at plosone@plos.org.

6. PLOS authors have the option to publish the peer review history of their article (what does this mean? ). If published, this will include your full peer review and any attached files.

**Do you want your identity to be public for this peer review?** For information about this choice, including consent withdrawal, please see our Privacy Policy .

Reviewer #1: No

Reviewer #2: No

Reviewer #3: No

---

## [Editor Report · Decision Letter 1]

Investigating the relationship between internal spinal alignment and back shape in patients with scoliosis using PCdare: a comparative, reliability and validation study

PONE-D-25-09339R1

Dear Dr. Kaiser,

We’re pleased to inform you that your manuscript has been judged scientifically suitable for publication and will be formally accepted for publication once it meets all outstanding technical requirements.

Kind regards,

Mohammad Amin Fraiwan

Academic Editor

PLOS ONE
---

## [Editor Report · Acceptance letter]

PONE-D-25-09339R1

PLOS ONE

Dear Dr. Kaiser,

I'm pleased to inform you that your manuscript has been deemed suitable for publication in PLOS ONE. Congratulations! Your manuscript is now being handed over to our production team.

Kind regards,

on behalf of

Dr. Mohammad Amin Fraiwan

Academic Editor

PLOS ONE